# Blue Light-Induced Mitochondrial Oxidative Damage Underlay Retinal Pigment Epithelial Cell Apoptosis

**DOI:** 10.3390/ijms252312619

**Published:** 2024-11-24

**Authors:** Mohamed Abdouh, Yunxi Chen, Alicia Goyeneche, Miguel N. Burnier

**Affiliations:** 1Cancer Research Program, Research Institute of the McGill University Health Centre, Montreal, QC H4A 3J1, Canada; yunxi.chen@mail.mcgill.ca (Y.C.); alicia.goyeneche@affiliate.mcgill.ca (A.G.); miguel.burnier@mcgill.ca (M.N.B.); 2The MUHC-McGill University Ocular Pathology & Translational Research Laboratory, McGill University, Montreal, QC H4A 3J1, Canada

**Keywords:** blue light, retinal pigment epithelial cells, oxidative stress, mitochondria damage, caspases activation, apoptosis, antioxidant

## Abstract

Reactive oxygen species (ROS) play a pivotal role in apoptosis. We reported that Blue Light (BL) induced oxidative stress in human retinal pigment epithelial (RPE) cells in vitro and increased drusen deposition and RPE cell apoptosis in human eyes. Here, we investigated the mechanisms underlying BL-induced damage to RPE cells. Cells were exposed to BL with or without the antioxidant N-acetylcysteine. Cells were analyzed for levels of ROS, proliferation, viability, and mitochondria membrane potential (ΔΨ_M_) fluctuation. We performed proteomic analyses to search for differentially expressed proteins. ROS levels increased following RPE cell exposure to BL. While ROS production did not affect RPE cell proliferation, it was accompanied by decreased ΔΨ_M_ and increased cell apoptosis due to the caspase cascade activation in a ROS-dependent manner. Proteomic analyses revealed that BL decreased the levels of ROS detoxifying enzymes in exposed cells. We conclude that BL-induced oxidative stress is cytotoxic to RPE cells. These findings bring new insights into the involvement of BL on RPE cell damage and its role in the progression of age-related macular degeneration. The use of antioxidants is an avenue to block or delay BL-mediated RPE cell apoptosis to counteract the disease progression.

## 1. Introduction

Age-related macular degeneration (AMD) is a progressive degenerative disease affecting the macula with subsequent irreversible vision loss [1,2,3]. Aside from genetic predisposing mutations, inflammation, smoking, and diet, the primary risk factor for disease development remains age, as it reaches 4% of individuals below the age of 50 but more than 27% of 80-year-old people [1,4]. Affected individuals suffer from considerable deterioration of their sharp central vision. This results from gradual failure of the Bruch’s membrane, choroidal capillaries, and retinal pigment epithelial (RPE) cells with ensuing dysfunction of photoreceptors [5,6]. On the other hand, AMD is linked to high socioeconomic issues, which constitutes a heavy charge on the health system regarding patients’ care [7,8].

AMD presents in wet form, which accounts for 15% of cases. It is due to growing capillaries that invade the subretinal space with subsequent leakage that induces hemorrhage. Patients in this cohort benefit from treatments that target neovascularization. The disease could also develop as a non-exudative dry form that concerns 85% of cases. This form develops following (i) drusen deposition underneath the macula between the Bruch’s membrane and the RPE layer and (ii) lipofuscin accumulation in RPE cells. For this form of the disease, there is no approved treatment, which requires a better understanding of the disease to identify new therapeutic targets [6,9,10,11,12,13,14].

Light was reported as another risk factor for the development and progression of AMD, mostly the dry form as RPE cell-stored lipofuscin, increased cell sensitivity to different spectra [1,15,16]. Ultraviolet (UV) and blue light (BL) wavelengths induce photooxidative stress and photochemical damage to exposed cells. Specifically, BL triggers RPE cell senescence and death [1,16,17,18,19,20,21,22,23,24,25]. In the eye, the cornea and the lens crystalline block UV spectra. Instead, the BL spectra diffuse through these structures, reach the retina and the underneath tissues, and affect their proper functioning [23]. During aging, the lens progressively turns yellowish, which buffers BL radiation entering the eye and protects the posterior eye structures [26,27]. Nonetheless, this naturally protective effect is lost during lens removal or replacement [28,29,30].

Oxidative stress and apoptosis are linked physiological phenomena. ROS and mitochondria play pivotal roles in the apoptotic cascade induction under both physiological and pathologic conditions [31,32,33]. In the context of AMD, we reported that BL induced oxidative stress and subsequent cytotoxicity to cultured human RPE cells and increased drusen deposition that triggered oxidative stress and RPE cell apoptosis in human cadaveric eye specimens [29,30,34,35]. The use of BL-filtering devices mitigated these effects. Here, we determined the mechanisms behind BL-induced damage to RPE cells. While increased ROS levels did not affect RPE cell proliferation, it induced a significant decrease in mitochondrial membrane potential and an increase in RPE cell apoptosis. BL-induced RPE cell apoptosis resulted from the activation of the caspase cascade in a ROS-dependent manner. The proteomic analyses revealed that BL decreased the expression levels of several ROS detoxifying enzymes in exposed RPE cells that will prolong the oxidative stress in these cells with a maintenance of the BL cytotoxic effects. Together, our findings bring new insights into the involvement of BL on RPE cell damage and its putative role in the progression of AMD. Filtering these radiations or the use of antioxidants are avenues to block or delay BL-mediated RPE cell apoptosis to counteract the disease progression.

## 2. Results

### 2.1. BL-Induced Oxidative Stress in RPE Cells

In this study, we used A2E-loaded ARPE-19 cells and human primary RPE cells (Appendix A) [34]. RPE cells were exposed to BL under a Solar Simulator to normalize in vitro light exposure to sunlight reaching the eye in vivo. We found that BL exposure significantly increased the levels of total cellular ROS and mitochondrial superoxide anion in both primary RPE cells and ARPE-19 cells (Figure 1 and Appendix A). These data show that BL induces oxidative stress in human RPE cells.

### 2.2. BL Is Cytotoxic to RPE Cells in a ROS-Dependent Manner

We reported that BL affected RPE cell growth, which might be due to an effect on cell proliferation or cell viability [34]. Here, we first investigated cell proliferation and found that BL did not affect RPE cell proliferation as assessed by cell cycle (Figure 2a and Appendix A). When we studied the effects of BL on cell viability, we found that it significantly increased RPE cell apoptosis, while it did not affect cell necrosis (Figure 2b and Appendix A). To determine whether BL-induced cytotoxic effects were linked to increased ROS production, we pretreated cells with the ROS scavenger NAC. We found that quenching ROS production abolished BL-induced apoptosis. This indicates that BL-elicited oxidative stress triggered apoptotic cell death in RPE cells.

### 2.3. BL Induces ΔΨ_M_ Collapse and Caspase Pathway Activation

It is well recognized that mitochondrial respiratory chain, oxidative stress, and cell growth are linked physiological processes [31,32,33]. Specifically, ROS production and ΔΨ_M_ defect drive apoptotic cascade induction. We determined the molecular mechanism links underlaying BL-induced oxidative stress and cytotoxic effects in RPE cells. We assessed ΔΨ_M_ and found that BL exposure significantly reduced it by 55% to 60% in primary RPE cells and ARPE-19 cells, respectively (Figure 3 and Appendix A). These effects were reduced following pre-treatment of cells with NAC.

As ΔΨ_M_ collapse is accompanied by caspase activation; we verified this apoptosis-inducing pathway. We observed that BL exposure increased by ~2–4 times the levels of activated caspases 9/3/7 (Figure 4 and Appendix A). Interestingly, all these effects were prevented following NAC pre-treatment of RPE cells, which stipulates that BL-induced oxidative stress in RPE cells triggers ΔΨ_M_ collapse and subsequent activation of caspase cascade-mediated apoptosis.

### 2.4. BL Decreased the Expression of ROS Detoxifying Enzymes in RPE Cells

To determine putative factors associated with BL effects on RPE cells, we performed proteomic analyses. We identified 2810 proteins, of which 1404 (50%) were detected in all analyzed RPE samples (Appendix A). High percentages of detected proteins were shared between non-treated cells (71.1–78.3%) and between BL-exposed cells (67.7–79.6%) (Figure 5a(i,ii)). In addition, non-exposed and BL-exposed cells shared 2369 proteins, while 288 and 153 proteins were exclusively present in one or other samples, respectively (Figure 5a (insert) and Appendix A). As a readout for the cell origin of analyzed proteins, we detected a panel of proteins that are specific markers of RPE (Table 1).

Notably, we found that 44 proteins were upregulated and 129 proteins were downregulated in BL-exposed cells (Appendix A). We focused our analyses on factors involved in cellular response to oxidative stress (Appendix A). We found that many ROS detoxifying enzymes were down-regulated in BL-exposed cells (Figure 5b,c).

To identify physiological processes to which the identified proteins are related, we clustered the most differentially expressed proteins into gene ontology categories. Characterization by biological process highlighted categories consistent with response to oxidative stress and cellular response to stress in proteins down-expressed in BL-treated cells (i.e., ETFDH, GSS, PXDN, and PRDX6; 5.3-fold decrease). In contrast, highly expressed in BL-treated cells clustered in categories consistent with apoptotic signaling and NHEJ-associated DNA repair pathways (i.e., ANXA5, HSPA5, PRKDC, THBS1, SLC25A5, SLC25A6 and TP53BP1; 32.4-fold enrichment) (Figure 6). This is in line with our findings that BL elicited ROS-mediated apoptosis and produced ROS-induced DNA damage that caused the activation of the DNA repair machinery.

## 3. Discussion

BL is a risk factor for AMD [15]. We reported that it induced oxidative stress in RPE cells in vitro [34] and increased drusen deposition that triggered RPE cell apoptosis in human eyes [29,30,35]. In this study, we determined molecular mechanisms underlying BL-induced damage to primary human RPE cells. Using a Solar Simulator, we normalized in vitro light exposure to light reaching the retina in vivo [36]. While this remains an artificial model and condition, it helps in understanding the behavior of RPE cells under sunlight-like illumination. BL increased ROS levels in RPE cells, eliciting a collapse in the ΔΨ_M_, and increasing apoptosis following caspase activation. Also, BL decreased detoxifying enzyme expression, which sustains oxidative stress and cytotoxic effects.

Light exposure is toxic to many tissues and underlies many diseases [37,38,39,40,41,42,43,44]. We reported on its involvement in the pathogenesis of uveal melanoma [45,46,47,48]. It is also responsible for other ocular diseases (cataract and AMD) [15,16,49]. While UV radiations are blocked by the cornea and lens, visible light crosses these tissues and reaches posterior eye structures [23,50,51]. Of these high-energy radiations, BL displays the most cytotoxic effects on RPE cells [16,17,18,19,20,21,22,24,25,52,53]. We deepen these observations using primary RPE cells from aged donors and by demonstrating a direct link between BL-induced ROS production and RPE cell cytotoxicity. The use of antioxidants rescued RPE cells from BL-induced damage (Figure 7). Notably, following BL exposure, the levels of cellular ROS increased more than seven times. Instead, the levels of the mitochondrial superoxide anion increased by only two times, due to the fact that this unstable intermediate is readily converted to more stable metabolites [54].

Increased ROS levels induce mitochondrial DNA damage and dysfunction, and subsequent cellular damage. This triggers various degenerative pathologies, such as AMD [55]. We found that BL induced ΔΨ_M_ collapse, activated caspase cascade, and caused cell apoptosis in a ROS-dependent manner. Therefore, mitochondrial dysfunction is likely to play an important role in the induction of the observed RPE cell apoptosis due to ROS production. Antioxidant mechanisms are suppressed in A2E-loaded RPE cells [16,56]. Our proteomic analyses showed that BL significantly reduced the levels of many antioxidant enzymes that might exacerbate RPE cell cytotoxicity. It should be highlighted that ROS may target different cellular components (i.e., proteins, lipids, DNA) to induce, for example, lipid peroxidation or DNA damage that culminate in cell dysfunctions.

During aging, RPE cells face different insults, such as light and oxidative stress. BL induces lipofuscin deposition and subretinal drusen accumulation [57,58]. During the visual cycle, these light-absorbing structures are processed by RPE cells. In order for photoreceptors to work effectively, outer segments need to be replaced daily, and RPE cells act as the recycling station for this phagocytosis-associated process [59,60]. That way, they ensure that debris does not build up underneath the retina. Phagocytized outer segments are digested in RPE lysosomes, but this reaction is hampered by oxidative stress. Subsequently, undigested residues form A2E-rich lipofuscin that has an absorbance peak at 350–435 nm. This increases RPE cell photosensitization and triggers a vicious circle [17,61,62,63,64,65]. In addition, deposition of drusen during aging causes a failure in the hydraulic conductivity and RPE cell malnutrition, with subsequent neurodegeneration [66]. We mimicked this situation in vitro by using ARPE-19 cells charged with synthetic A2E. We found that BL effects on these cells were almost the same as on primary cells.

Based on our findings, many therapeutic avenues for AMD are possible. Interventions that counteract oxidative stress were shown to be beneficial in the treatment of many diseases [67,68,69,70]. Application of this strategy is promising, as antioxidants are already used in ophthalmologic clinics. Alteration of mitochondrial functions suggests that it may be a potential target for disease prevention. Mitochondrial stimulation protects RPE cells from oxidative damage [67,68,69,70,71]. Following cataract surgery, the protective function against BL of the age-associated yellowing lens is lost [26,27]. The recent use of the BL-filtering intraocular lenses to replace the natural lens crystalline was reported to restore this deficiency by reducing the levels of produced ROS and RPE cell mortality [34,72,73]. These devices filter the “bad” BL (below 460 nm) but not the “good” BL (above 460 nm) that is involved in the regulation of the circadian rhythm [19,74,75]. Photobiomodulation (PBM), a process that regulates physiological conditions following light exposure, promotes cellular fitness. PBM is currently used in physiotherapy, arthritis, wound repair, and sports medicine [76,77,78]. It acts through the activation of the mitochondrial respiratory chain with subsequent normalization of cellular functions (i.e., proliferation, survival, and cytoprotection) [76]. Recently, PBM was shown to have beneficial effects for AMD as it induced a reduction in the size and number of Drusen [79,80,81]. Its action could also pass through the regulation of oxidative stress and mitochondrial function at the level of RPE cells.

## 4. Materials and Methods

### 4.1. Human Eye Procurement for Primary RPE Cell Isolation and Cell Culture

Human eyes (*n* = 6, 3 males and 3 females, 65–76-years-old) were obtained from the Centre Hospitalier Universitaire de Québec (Canada), following informed consent from the donor’s next of kin, and were used in accordance with a protocol approved by the ethic board of the RI-MUHC (#2019–5314) and with The Code of Ethics of the World Medical Association.

Primary RPE cell cultures were established as reported previously [34,82]. In all experiments, cells were used between the second and fourth passages (exponential growth phase and presence of cytoplasmic pigmented granules) (Appendix A) [83]. ARPE-19 cell line was obtained from Cedarlane (ON., Canada) and was maintained in DMEM-F12 medium supplemented with 10% FBS and antibiotics (Corning, AZ, USA). These cells were used for all experiments at early passages (˂20). As reported previously, ARPE-19 cells were loaded with A2E (20 µM) 24 h before exposure to light [34].

### 4.2. Cell Exposure to BL

Cells were exposed to BL when they reached 70% confluence. Cultures were maintained in the dark wrapped in aluminum foil at 37 °C and 5% CO_2_. Cell culture medium was removed, replaced with D-PBS supplemented with calcium and magnesium, and cells were exposed under a solar simulator (TSS-156R, OAI, OAInet, Milpitas, CA, USA) set at 30 mW/cm^2^ for 30 min in the presence or absence of NAC. NAC (1 mM; Sigma-Aldrich, St. Louis, MO, USA) was added to cells 24 h before exposure to BL and during BL exposure. A Blue Dichroic Filter (Edmund Optics Inc., Bengaluru, India) was used to allow only BL to pass and reach cells (Appendix A).

### 4.3. Reactive Oxygen Species (ROS) Detection

We analyzed both total cellular ROS and mitochondrial superoxide anions using DCF-DA and MitoSox Red probes, respectively, according to manufacturer protocols (ThermoFisher, Waltham, MA, USA). Fluorescence was read using an Infinite M200 Pro microplate reader (Tecan, Mennedorf, Switzerland). Reading parameters were introduced manually to normalize fluorescence measurements between experiments.

### 4.4. Cell Cycle and Apoptosis Analyses

For cell cycle analyses, cells were fixed in ice-cold ethanol (70%) for 2 h and labeled with propidium iodide (PI (50 μg/mL); Sigma Aldrich, St. Louis, MO, USA). Cells were acquired in a BD FACSCanto II flow cytometer at ~400 events/second flow rate. Doublets were excluded by creating a combination of FSC-channel bivariate plots using Area vs. Height parameters. For apoptosis analysis, we used the Alexa Fluor 488 Annexin V/Dead Cell Kit (ThermoFisher, Waltham, MA, USA) following the manufacturer’s instructions. ~20.000 cells were acquired per sample at ~500 events/second rate. Analyses were performed using FlowJo software (version 10.10) [84,85].

### 4.5. Mitochondria Membrane Potential (ΔΨ_M_) Measurement

ΔΨ_M_ was assessed using the JC-1 probe according to the manufacturer’s instructions (Cayman, MI, USA). Fluorescence was read using the Infinite M200 Pro microplate reader. Two measures were performed at Ex/Em (535/595 nm and 485/535 nm) for red J-aggregates and green monomers, respectively. Data were presented as ratio of J-aggregates to monomer values.

### 4.6. Western Blot and Mass Spectrometry (MS) Proteomic Analyses, and Database Search

For MS analyses, cell samples were resuspended in PBS. Cell preparations for Western Blot were homogenized in RIPA containing protease inhibitors (Sigma Aldrich, St. Louis, MO, USA) at 4°C for 30 min.

For immunoblotting, proteins were resolved in precast polyacrylamide gel and transferred to PVDF membranes (Bio-Rad, Hercules, CA, USA). Membranes were probed with rabbit anti-caspase 9 (cleaved Asp353) (ThermoFisher, Waltham, MA, USA) and mouse anti-β-actin (Sigma Aldrich, St. Louis, MO, USA) antibodies, followed by HRP-conjugated goat anti-rabbit and goat anti-mouse antibodies (Sigma Aldrich, Waltham, MA, USA). Protein signals were visualized using ECL prime Western Blot detection (Sigma Aldrich, St. Louis, MO, USA) in a ChemiDoc System (BioRad, Hercules, CA, USA). Densitometric analysis was performed using ImageJ software (version 1.54g).

Liquid chromatography-tandem mass spectrometry proteomic analyses were performed on protein samples as previously described [86,87]. Raw data were converted into *.mgf format (Mascot generic format) to use the Mascot2.6.2 search engine to search against human protein sequences (Uniprot 2019). Database search results were loaded onto Scaffold 4.10.0 for spectral counting, statistical treatment, data visualization and quantification. Samples with low total protein counts and low spectrum counts were excluded from the analyses. A *p*-value cut-off of 0.05 and a fold-value change of ≥2 were used to identify the differentially expressed proteins. The identified protein list in Scaffold was exported to Microsoft Excel sheets and uploaded into the DAVID Bioinformatics database (v2023q4) for the analysis of functional gene enrichment and annotation (gene ontology analyses). In addition, bioinformatic analyses were performed using the FunRich software (version 3.1.3).

### 4.7. Caspases 3/7 Activation Analyses

Caspase pathway activation was analyzed using the CellEvent Caspase-3/7 probe (ThermoFisher, Waltham, MA, USA) as per the manufacturer’s protocol. Following staining, cells were mounted with coverslips in Mounting Medium with DAPI (Vectorlabs, MA, USA) and visualized using an LSM780 confocal microscope (Zeiss, Jena, Germany).

### 4.8. Statistical Analyses

All experiments were performed with 6 independent primary RPE cell cultures or at least 3 independent ARPE-19 cell cultures. Data were compared using an ANOVA followed by the Dunnett post hoc test for multiple comparisons with one control group. A *p* value < 0.05 was considered statistically significant.

## 5. Conclusions

BL exposure elicits oxidative stress in RPE cells that triggers mitochondrial damage and cell apoptosis. Quenching ROS produced following BL exposure provides protective effects to these cells. Proposed strategies for counteracting the deleterious effects of BL boil down to blocking these radiations or targeting their downstream cellular effects. Overall, our findings give a rationale for the use of multiple strategies to prevent the eye’s posterior segment and mainly the RPE layer from BL deleterious effects.

## Figures and Tables

**Figure 1 ijms-25-12619-f001:**
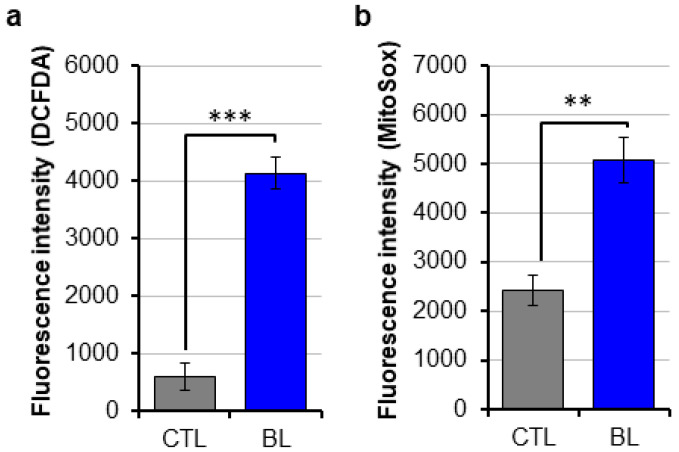
BL-induced oxidative stress in primary RPE cells. Primary human RPE cells were exposed to BL for 30 min. (**a**) Cells were analyzed for the production of total cellular ROS using the DCF-DA probe. (**b**) Cells were analyzed for the production of mitochondrial superoxide anion using the MitoSox Red probe. Data are presented as mean ± SD (*n* = 6 independent experiments each repeated in quadruplicates, ** *p* ˂ 0.01, *** *p* ˂ 0.001).

**Figure 2 ijms-25-12619-f002:**
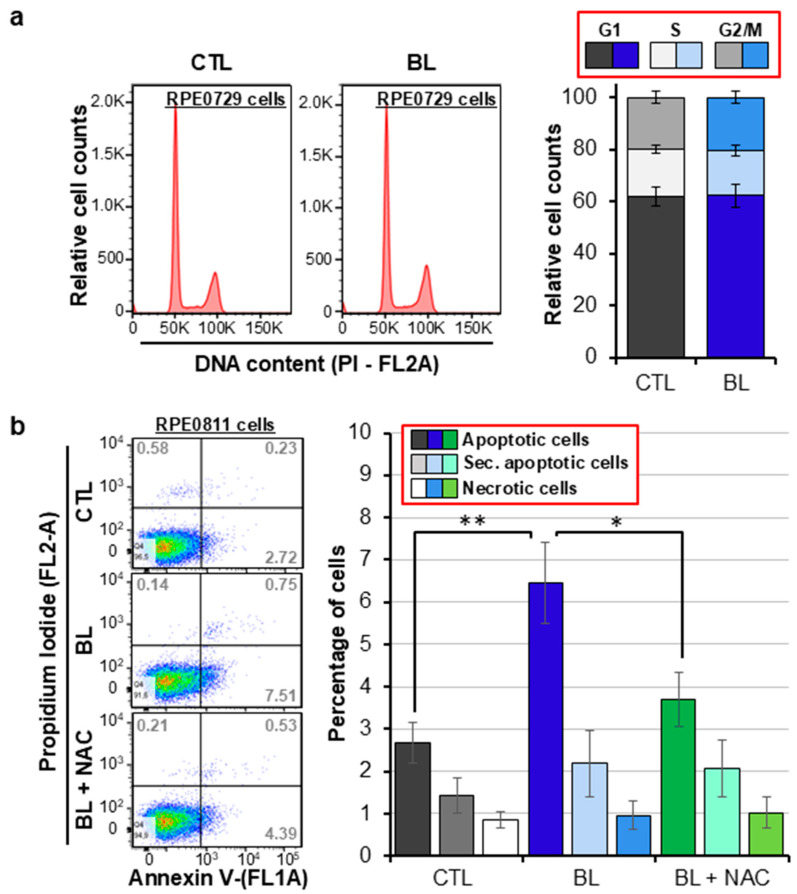
BL-induced RPE cells apoptosis in a ROS-dependent manner. Primary human RPE cells were exposed to BL for 30 min. (**a**) 24 h post-BL exposure, cells were labeled with propidium iodide (PI) and analyzed for their proliferation. Representative cell cycle phase distribution histograms are shown where the first peak (at 50 k) corresponds to cells in the G1 phase, the second peak (at 100 k) corresponds to cells in the G2/M, and area in between the peaks represents cells in the S phase. The graph displays the percentages of cells in these different phases of cell cycle as analyzed using FlowJo software (v10.10). (**b**) 6 h post-BL exposure, cells were labeled with Annexin V and PI and analyzed for the percentages of apoptotic cells by flow cytometry. Representative Annexin V/PI density plots are shown that display live cells (left-lower quadrants), apoptotic cells (right-lower quadrant), secondary apoptotic cells (right-upper quadrant) and necrotic cells (left-upper quandrant). The numbers represent the percentages of cells in the respective quadrants as analyzed using FlowJo software. The graph displays means of the percentages of primary apoptotic, secondary apoptotic and necrotic cells. Data are presented as mean ± SD (*n* = 6 independent experiments, * *p* < 0.05, ** *p* ˂ 0.01).

**Figure 3 ijms-25-12619-f003:**
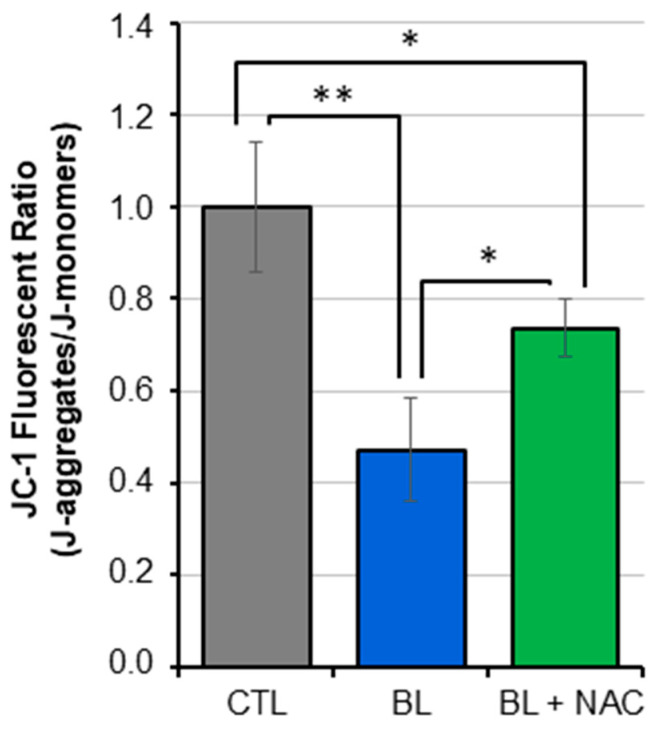
BL reduced the mitochondrial membrane potential in a ROS-dependent manner. Primary human RPE cells were exposed to BL for 30 min, and cells were stained with JC-1 probe. Fluorescence of J-aggregates and J-monomers were measured. Data are expressed as the ratio between the 2 measures and are presented as mean ± SD (*n* = 6 independent experiments each repeated in quadruplicates, * *p* < 0.05, ** *p* ˂ 0.01.

**Figure 4 ijms-25-12619-f004:**
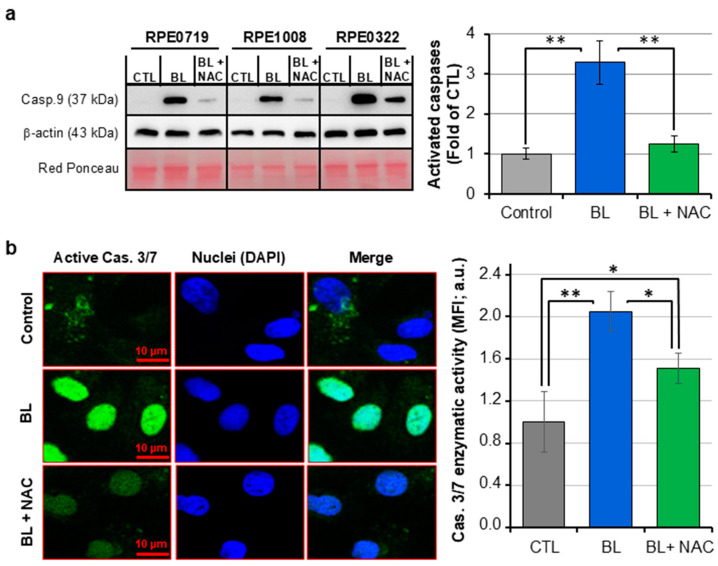
BL increased caspase cascade activation in ROS-dependent manner. Primary human RPE cells were exposed to BL for 30 min. (**a**) Proteins extracts were analyzed by immunoblot for the activation of Caspase 9. β-actin and red ponceau staining were used as calibrators for proteins loading. The graph shows the levels of caspases activation in the corresponding samples. Data are expressed as the densitometer values relative to the value in control sample set at 1. (**b**) Cells were loaded with CellEvent Caspase 3/7 Green. Pictures were acquired using a LSM780 confocal microscope. The graph displays the levels of caspases 3 and 7 activation in the corresponding samples. Data are expressed as mean fluorescence intensity (MFI) measured in an Infinite M200 Pro microplate reader relative to the value in control sample set at 1. Data are presented as mean ± SD (*n* = 3–5 independent RPE cells samples). * *p* < 0.05, ** *p* < 0.01.

**Figure 5 ijms-25-12619-f005:**
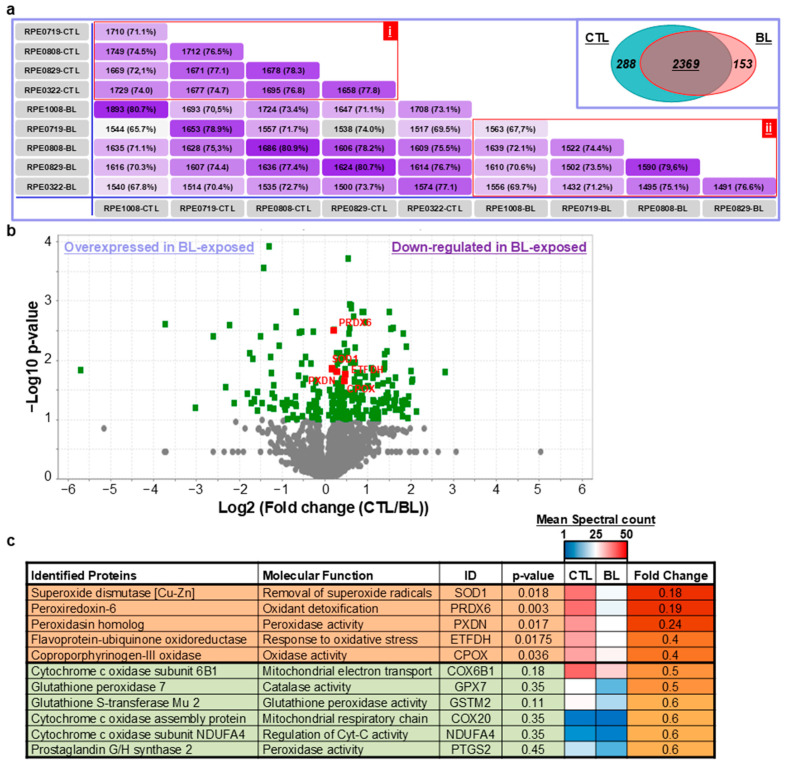
BL decreased the expression levels of ROS detoxifying enzymes. Primary human RPE cells were exposed to BL for 30 min. (**a**) Venn diagram analyses. Samples datasets were compared for shared proteins between non-exposed cells (CTL), between BL-exposed cells (BL), and between exposed and non-exposed cells. In the insert, CTL and BL-exposed RPE cells shared 2369 proteins, while 288 and 153 proteins were exclusively present in CTL and BL-exposed RPE cells, respectively (see Appendix A for the full list of proteins). (**b**) Volcano plot representation of proteins significantly and differentially expressed between CTL and BL-exposed RPE cells (See Appendix A for the respective protein lists). For statistical analyses, we set the analysis for a T-Test with a significant level at 0.05). ■: Significant, ●: Nonsignificant. (**c**) Table showing a short list of ROS detoxifying enzymes which expression is decreased in BL-exposed cells. In Orange lines are shown differentially and significantly expressed proteins. In Green are shown proteins with decreased expression levels in BL-exposed cells but not reaching the statistical significance (See Appendix A for the full list of identified proteins). Data are obtained from the analysis of 5 RPE cell lines derived from 5 different eye donors.

**Figure 6 ijms-25-12619-f006:**
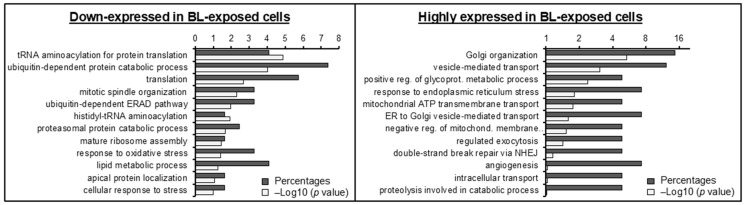
Gene ontology classification of proteomic data for differentially expressed proteins in primary RPE cells exposed or not to BL. The most enriched categories in biological processes, as analyzed by the DAVID bioinformatics platform, are shown. Data were collected from protein preparations obtained from five patient-derived RPE cells.

**Figure 7 ijms-25-12619-f007:**
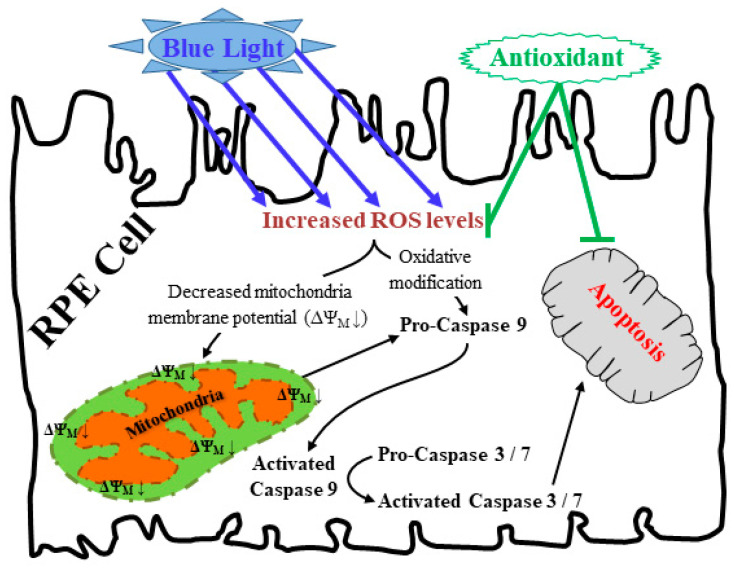
BL-induced RPE cell apoptosis model. BL-mediated oxidative stress triggered mitochondrial damage and concomitant caspase cascade activation. These effects induced RPE cell apoptosis, that could be reversed by antioxidants (i.e., N-acetyl cysteine; NAC).

**Table 1 ijms-25-12619-t001:** List of read-out proteins in RPE cells analyzed samples.

Identified Proteins	Alternate ID	Molecular Function
Keratin 8	KRT8	Scaffold protein binding
Keratin 18	KRT18	Scaffold protein binding
Tight junction protein -1	ZO-1	Cell–cell adhesion, tight junction
Tight junction protein-2	ZO-2	Cell–cell adhesion, tight junction
Retinal dehydrogenase 1	ALDH1A1	Retinol metabolic process
Cell retinoic acid-binding protein 2	CRABP2	Retinoic acid biosynthetic process
All-trans-retinol 13,14-reductase	RETSAT	Retinol metabolic process
Retinol dehydrogenase 11	RDH11	Retinol metabolic process
Retinoid-inducible carboxypeptidase	SCPEP1	Retinoic acid metabolic process
All-trans-retinol dehydrogenase [NAD]	ADH1B	Retinoic acid metabolic process
Retinol dehydrogenase 14	RDH14	NADP-retinol dehydrogenase activity
Plasma membrane calcium-transporting	ATP2B4	Aging, neural retina development
Plasma membrane calcium-transporting	ATP2B1	Aging, neural retina development
Solute carrier family 2	SLC2A1	Photoreceptor cell maintenance
Lysosomal protective protein	CTSA	Chaperone-mediated autophagy
Lysosome membrane glycoprotein	LAMP2	Protein complex assembly
Lysosomal Pro-X carboxypeptidase	PRCP	Energy/glucose homeostasis
Lysosomal acid phosphatase	ACP2	Lysosome organization
Lysosomal alpha-mannosidase	MAN2B1	Cell proteins modification process

## Data Availability

Research data will be shared upon formal request.

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
