# Peer review of "Blue Light-Induced Mitochondrial Oxidative Damage Underlay Retinal Pigment Epithelial Cell Apoptosis"

_ijms, 2024, doi:10.3390/ijms252312619_

Round 1

Reviewer 1 Report

Comments and Suggestions for Authors

Blue light-induced mitochondrial oxidative damage underlay retinal pigment epithelial cell apoptosis

Age-related macular degeneration (AMD) is a progressive degenerative disease af- fecting the macula with subsequent irreversible vision loss [1-3]. Aside from genetic predisposing mutations, inflammation, smoking and diet, the primary risk factor for disease development remains age as it reaches 4% of individuals below the age of 50 but more than 27% of 80-year-old people [1,4]. Affected individuals suffer from considerable deterioration of their sharp central vision. This results from gradual failure of the Bruch's membrane, choroidal capillaries and retinal pigment epithelial (RPE) cells with ensuing dysfunction of photoreceptors [5,6]. On the other hand, AMD is linked to high socioeconomic issues which constitutes a heavy charge on health system regarding patients’ care.

The topic presented by the authors is very interesting and very worrying, especially today there are many patients who suffer and do not know what to do and I believe. That this type of work must exist to solve this disease.

After reading the complete manuscript, I have observed that the literature consulted for this manuscript is well updated.

In relation to these observations, I have a few questions:

1.    Why do you not use hydrogen peroxide (H2O2) in your experiments?

2.    Explain the mechanisms behind BL-induced damage to RPE cells and how the use of BL filtering devices mitigated these effects?

3.    Explain what are the mechanisms underlying the damage induced by Blue Light (BL) in RPE cells?

4.    How do you explain the involvement of Blue Light (BL) in RPE cell damage and its role in the progression of Age-Related Macular Degeneration?

5.    What role does the antioxidant N-acetylcysteine​​(NAC) play in this study? Is it a way to block or delay Blue Light (BL)-mediated RPE cell apoptosis to counteract the progression of age-related macular degeneration?

Author Response

Reviewer #1:

  1. Why do you not use hydrogen peroxide (H2O2) in your experiments? We thank the reviewer for rising this important idea. It is known that H2O2 is damaging to RPE cells (e.g., Tang et al. Journal of Controlled Release 357 (2023): 356-370). In future experiments, we plan to use this strategy to induce cellular oxidative stress in RPE cells to analyze the underlying mechanisms responsible for the cytotoxic effects. However, in our present study, our main goal is to determine the effects of blue light as inducer of oxidative damage by analyzing both mitochondrial and total cellular ROS.

  1. Explain the mechanisms behind BL-induced damage to RPE cells and how the use of BL filtering devices mitigated these effects? As shown by our data, two mechanisms are responsible for the damaging effects of BL on RPE cells: (i) the increase of ROS levels as mentioned in the Discussion (Lines 206 to 212 and Figure 7), and (ii) the reduction in the levels of antioxidant enzymes (Proteomic data). Together this exacerbates the BL-induced oxidative stress. It is known that ROS may target different cellular components (proteins, lipids, DNA) to induce, for example, lipid peroxidation or DNA damage that culminate in cell dysfunctions. This is now added in the Discussion (Lines 224 to 226: ‘’It should be highlighted that ROS may … in cell dysfunctions.’’). Regarding how the use of BL filtering devices mitigated BL effects on RPE cells, we highlighted this in the Discussion (Lines 246 to 250) based on our previously published data (reference 34) and those of others (references 78 and 79).

  1. Explain what are the mechanisms underlying the damage induced by Blue Light (BL) in RPE cells? We sincerely think that the answer to this question was addressed as mentioned in the Point 2 (please see Lines 224 to 226 in the Discussion).

  1. How do you explain the involvement of Blue Light (BL) in RPE cell damage and its role in the progression of Age-Related Macular Degeneration? We honestly believe that we addressed this point in the discussion (Lines 227 to 239). We built this explicative model based on our data and those of other laboratories as mentioned at the beginning of the Discussion (Lines 192 to 194: ‘’BL is a risk factor for AMD [15]. … apoptosis in human eyes [29,30,35].’’).

  1. What role does the antioxidant N-acetylcysteine (NAC) play in this study? Is it a way to block or delay Blue Light (BL)-mediated RPE cell apoptosis to counteract the progression of age-related macular degeneration? We stated that the use of antioxidants is an avenue to block or delay BL-mediated RPE cell apoptosis to counteract the disease progression. We believe that this will depend on the disease status, meaning that if the disease is detected precociously, the use of antioxidants will be more efficient in blocking cell apoptosis, but in the contrary, if the disease had already severely progressed, the use of antioxidants will be expected to delay cell apoptosis.

Reviewer 2 Report

Comments and Suggestions for Authors

This is a straightforward and clearly reported analysis of the effects of blue light on A2E loaded cultured ARPE19 cells and primary human RPE cells. The authors evalutated the effects of blue light exposure on reactive oxygen species, cell replication, apoptosis, mitochondrial membrane potential and the beneficial effects of N-acetyl cysteine. This is an important set of findings that support previous findings from this lab and their interpretation. I have only a couple of suggestions to improve this manuscript. 

1.        Please include a transmission spectrum of the blue dichroic filter that was used to filter the light from the solar simulator. 

2.        Fig. 2B. The authors should describe in the figure legend how the data in the left panel was used to calculate the data in the right panel. The numbers in the panels of Fig. 2A also need to be defined.

Author Response

Reviewer #2.

  1. Please include a transmission spectrum of the blue dichroic filter that was used to filter the light from the solar simulator. This information is now added in the Supplementary Figure 1 as (Panel B), and mentioned in the manuscript (Line 280).

2. Fig. 2B. The authors should describe in the figure legend how the data in the left panel was used to calculate the data in the right panel. The numbers in the panels of Fig. 2A also need to be defined. This is done accordingly in the legend to Figure 2.
